# Reduction in social learning and increased policy uncertainty about harmful intent is associated with pre-existing paranoid beliefs: Evidence from modelling a modified serial dictator game

**Joseph M. Barnby**[1,2]*, **Vaughan Bell**[1,3], **Mitul A. Mehta**[1,2], **Michael Moutoussis**[4,5]

**1** Cultural and Social Neuroscience Group, Department of Neuroimaging, Institute of Psychiatry, Psychology & Neuroscience, King's College London, London, United Kingdom, **2** Neuropharmacology Group, Department of Neuroimaging, Institute of Psychiatry, Psychology & Neuroscience, King's College London, London, United Kingdom, **3** Research Department of Clinical, Educational, and Health Psychology, University College London, London, United Kingdom, **4** Wellcome Centre for Human Neuroimaging, University College London, London, United Kingdom, **5** Max-Planck–UCL Centre for Computational Psychiatry and Ageing, University College London, London, United Kingdom

* joe.barnby@kcl.ac.uk

**Data Availability Statement:** All data and analysis scripts are freely available on the Open Science

## Abstract

Current computational models suggest that paranoia may be explained by stronger higher-order beliefs about others and increased sensitivity to environments. However, it is unclear whether this applies to social contexts, and whether it is specific to harmful intent attributions, the live expression of paranoia. We sought to fill this gap by fitting a computational model to data (n = 1754) from a modified serial dictator game, to explore whether pre-existing paranoia could be accounted by specific alterations to cognitive parameters characterising harmful intent attributions. We constructed a 'Bayesian brain' model of others' intent, which we fitted to harmful intent and self-interest attributions made over 18 trials, across three different partners. We found that pre-existing paranoia was associated with greater uncertainty about other's actions. It moderated the relationship between learning rates and harmful intent attributions, making harmful intent attributions less reliant on prior interactions. Overall, the magnitude of harmful intent attributions was directly related to their uncertainty, and importantly, the opposite was true for self-interest attributions. Our results explain how pre-existing paranoia may be the result of an increased need to attend to immediate experiences in determining intentional threat, at the expense of what is already known, and more broadly, they suggest that environments that induce greater probabilities of harmful intent attributions may also induce states of uncertainty, potentially as an adaptive mechanism to better detect threatening others. Importantly, we suggest that if paranoia were able to be explained exclusively by core domain-general alterations we would not observe differential parameter estimates underlying harmful-intent and self-interest attributions.

Framework at the following address: https://osf.io/24urf/.

**Funding:** JMB is supported by the UK Medical Research Council (MR/N013700/1) and King's College London member of the MRC Doctoral Training Partnership in Biomedical Sciences. The Max Planck – UCL Centre for Computational Psychiatry and Ageing is a joint initiative of the Max Planck Society and UCL. MM is supported by the Wellcome Trust as a member of the 'Neuroscience in Psychiatry Project' (NSPN) which is funded by a Wellcome Strategic Award (ref 095844/7/11/Z). MM also receives support from the NIHR UCLH Biomedical Research Centre. The funders had no role in study design, data collection and analysis, decision to publish, or preparation of the manuscript.

**Competing interests:** The authors have declared that no competing interests exist.

## Author summary

A great deal of work has tried to explain paranoia through general cognitive principles, although relatively little has tried to understand whether paranoia may be explained by specific changes to social learning processes. This question is crucial, as paranoia is inherently a social phenomenon, and requires mechanistic explanations to match with its dynamic phenomenology. In this paper we wanted to test whether pre-existing and live paranoid beliefs about others specifically altered how an individual attributed harmful intent–the live expression of paranoia–to partners over a series of live interactions. To do this we applied a novel computational model and network analysis to behavioural data from a large sample of participants in the general population that had played a modified Dictator game online, and required them to attribute whether the behaviour of their partner was due to their intent to harm, or their self-interest, on two mutually exclusive scales. Pre-existing paranoid beliefs about others reduced the value of new partner behaviours on evolving attributions of harmful intent. We suggest that both pre-existing paranoid beliefs and momentary paranoia may incur an adaptive cognitive state to better track potentially threatening others, and demonstrate phenomenological specificity associated with mechanisms of live paranoia.

## Introduction

Paranoia is the unwarranted belief that others intend to do us harm [1]. Paranoid beliefs are associated with a range of factors, including psychotic disorders [2,3] recreational drugs [4], sleep deprivation [5–7], epilepsy [8], and acute stress [9]. Phenomenologically, paranoia exists as a continuum in the general population ranging from fleeting thoughts to frank paranoid delusions [10]. Cognitive studies of paranoia have suggested a role for changes in analytic reasoning and belief flexibility [11], a jumping-to-conclusions probabilistic reasoning style [12], greater sensitivity to task reversals [13], and altered reward learning [14].

Social experiments using interactive game theory tasks have found that paranoia reduces the threshold for attributing harmful intent in ambiguous social exchanges [15–18]. More recently, Barnby et al. [19] extended this work and additionally found that individuals high in paranoia were more likely to reduce high harmful intent attributions after a higher initial peak when interacting with partners who were consistently fair. This finding potentially suggested increased uncertainty of live paranoid social inferences in those with higher baseline pre-existing paranoid beliefs. However, other experimental results have been mixed with regard to social inferences—Wellstein et al. [20] reported reduced flexibility in social sensitivity when making advice decisions.

Explanations for paranoia have increasingly focused on the role of learning and have been modelled based on the hypothesis that the brain processes uncertain information and appropriately revises beliefs depending on their baseline certainty and the impact of observations. This is the 'Bayesian Brain hypothesis', as appropriate revision of probabilistic beliefs is given by Bayes' rule. In these models, predictions or 'beliefs' regarding the environment are generated and updated based on incoming prediction errors. These beliefs are putatively organised into linked hierarchies ranging from simple predictions regarding sensory data to increasingly abstract predictions that encode high-level features of the environment. Prediction errors are weighted to signify their reliability (or 'precision') which determines their influence in updating beliefs. Converging evidence from Bayesian modelling of responses in paranoia [21,13] and schizophrenia [22] suggest that paranoia is potentially driven by expectations of high

environmental volatility and higher sensitivity to environmental changes that are not sufficiently updated by previous interactions within a task.

This raises a number of specific questions: whether these findings apply to social learning in paranoia, to what extent they are specific to social threat attributions, and how they interact with levels of pre-existing paranoia. Specifically, we wanted to assess whether pre-existing paranoia was associated with greater social belief uncertainty and less influence of prior partner interactions on intentional harm.

To this end, we built a computational model of participants' trial-by-trial variability of harmful intent and self-interest attributions based on a fully normative role of belief uncertainty. Beliefs here are not necessarily beliefs in the form of declarative propositional attitudes ("I believe Paris is the capital of France") but representations encoding probability distributions, necessary for performing approximate probabilistic information processing. Using data from a previous study [19] participants completed a measure of paranoia and were subsequently paired with an opponent for a six-round modified Dictator Game. Here, the dictator decides how to distribute a sum of money which is split between the dictator and the receiver, which the receiver must accept. The true motivation for the decision is ambiguous but receivers rate the extent to which the decision has been motivated by harmful intent (a threat-related attribution about the partner) and the extent to which it is motivated by self-interest (a non-threat-related attribution about the partner) for each economic exchange. In this study, all receivers are paired with fair, unfair, and partially fair dictators for six rounds each in a counter-balanced fashion.

We applied the model to estimate initial attributional strength (pHI0, pSI0) and uncertainty (uHI0, uSI0), in addition to how informative a partner's behaviours were in regard to a change in intentions (u$\Pi$) and how often attributions were updated from one dictator to the next ($\eta$). We thus used a spectrum of parameters describing uncertainty of different expectations. For example, a participant might be quite certain that others have high harmful intent (low uHI0) but be very uncertain about their partner's actions (policy or strategy), reflected in a high value of u$\Pi$. This detailed parameterization then allowed us to use the population variation of these parameters to infer whether a more limited range of cognitive styles in fact obtains, for example whether uncertainty into all types of beliefs vary together. The key parts of the model are described in Table 1 and full details of model development are given in Methods.

Before examining questions of interest, we validated the model predictively. We fitted parameter values to each participant and used these to create synthetic data. We then tested model validity by examining if synthetic data reproduced behavioural results which were not knowingly 'designed into' the model. All data and code are available online (https://osf.io/24urf/).

We then used measures from the model to test whether each uncertainty parameter increased with pre-existing paranoid beliefs. Second, we tested whether harmful intent attributions about the partner depended on pre-existing paranoia (as measured via the Green Paranoid Thoughts Scale; GPTS, [23]). Third, we tested if higher paranoia predicted how influential each decision by a partner was on altering social inferences. Finally, we used network modelling to understand the relationship between latent parameters, and the moderation of these relationships by pre-existing paranoia.

We expected pre-existing paranoia to lead to greater initial attributional uncertainties, greater initial attributional strength, a reduced influence of previous partner interactions, and less consistency between partner attributes and observed behaviours.

**Table 1. Glossary of model terms and model description.**

| Model measure | Technical definition and abbreviation | Key roles |
|---|---|---|
| Baseline level of Harmful Intent attribution | Mode of prior probability distribution of harm intent, $pHI_0$. It is the starting value of the mode $pHI_t$. | Greater $pHI_0$ leads to greater attributions of intent to harm initially, but how persistent this is depending both on evidence encountered (Dictator decisions seen) and, crucially, the uncertainty (inverse-strength) with which this belief is held. |
| Baseline uncertainty of Harmful Intent attribution | Spread of prior probability distribution of harm intent, $uHI_0$. It is the starting value of the uncertainty $uHI_t$. | Greater $uHI_0$ denotes reduced confidence about attributions of harmful intent, and more willingness to believe that a Dictator who acts less generously than expected has higher intent to harm. The balance of uncertainty about Harm intent, $uHI_t$, and uncertainty about Selfish intent, $uSI_t$, determines the balance of which attribute is updated more on the basis of the dictator's observed behaviour. Greater uncertainty $uHI_t$ directly contributes to greater variability in harmful intent attributions. |
| Baseline level of Self-Interest attribution | Mode of prior probability distribution of selfish intent, $pSI_0$; the starting value of the mode $pSI_t$. | Exactly analogous to $pHI_0$ above. |
| Baseline uncertainty of Self-Interest attribution | Spread of prior probability distribution of selfish intent, $uSI_0$; starting value of the uncertainty $uSI_t$. | Exactly analogous to $uHI_0$ above. |
| Partner policy uncertainty | Uncertainty parameter $u\Pi$ through which partner attributes are believed to lead to observed behaviours. | Unlike other uncertainties, this is not a spread of the distribution of both HI and SI attributions. The higher this uncertainty value, $u\Pi$, the less informative each observed return by the dictator is. Low $u\Pi$ means that one can be certain that the actions of the dictator were not 'by chance', but due to their true attributes. |
| Learning rate, a.k.a. belief-update parameter, from one dictator to the next | Weight $\eta$ by which the prior belief distribution over partner attributes shifts toward the distribution posterior to observing Dictator behaviour | A higher $\eta$ leads the starting assumptions of dictators after the first one seen to be influenced by prior dictator behaviour seen so far. It can be thought of as a strength of belief that the Dictators seen during the experiment will resemble each other. |
| Model fit | Log-posterior probability $lp$ that the fitted parameters gave rise to the data for this participant | A high $lp$ means that if given the fitted parameters, the model would closely reproduce the attributions of the participant. Note that a bad fit might be because the participant is behaving erratically (e.g. because of a high $u\Pi$) or that their pattern of behaviour is consistent in its own terms, but not captured by the model (e.g. a 'magical thinking' participant that alternates between two values in consecutive trials). |

## Result

All reported statistics are beta-coefficients of the top model following model averaging unless otherwise stated, and effect sizes (ES) are stated if they differ from the beta-coefficient.

## Model fitting

Our model broadly aims to capture the ideal Bayesian inference given the possible decision policies of a partner–from fair to unfair, selfish to harmful. We initially checked that the model could detect inferences better than chance (log likelihood = -4.394). The log likelihood that the model fitted the data were calculated across trials, dictators and divisions of the GPTS. Modelling fitting was adequate–for each trial, dictator and division of paranoia, log likelihood values staying much above -4.394, the value corresponding to a model 'hedging' its prediction equally over all possible participant responses (see Fig 1). Linear mixed model analyses using participant identification (ID) as a random term suggest that as trials progressed overall from one to eighteen (-0.012, 95%CI: -0.021, -0.003; ES = -0.04) and at higher values of the GPTS (-0.005, 95%CI: -0.007, -0.003; ES = -0.06) participants were less able to be predicted by our model,

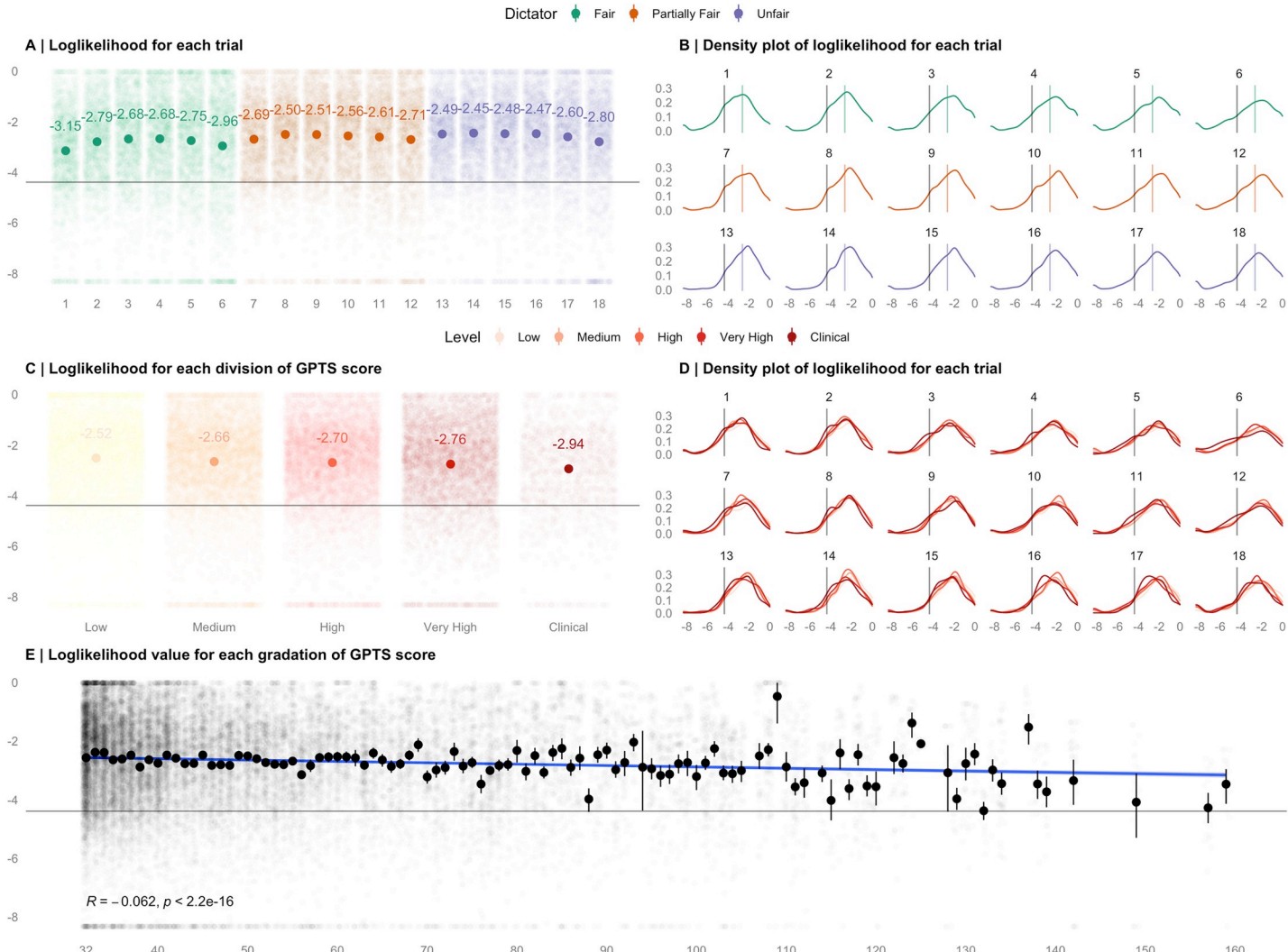

**Fig 1. Log Likelihood values of the model. (A)** Computed mean log likelihoods for each trial, coloured by dictator type. **(B)** Density plots of log likelihood values for each trial, coloured by dictator type. Coloured lines represent group means **(C)** Computed mean log likelihoods for each GPTS score quantile and clinical score cut off (Green et al., 2008). **(D)** Density plots of log likelihood values for each trial across each GPTS score quantile and clinical score cut off (Green et al., 2008). **(E)** The association between GPTS scores (minimum score = 32) and loglikelihood values. Dots = mean loglikelihood value across that score of the GPTS. Lines = 95% confidence intervals. The grey line in each plot (at -4.394) represents the loglikelihood that would be observed if the model was capturing behaviour by chance.

whereas our model was better able to predict behaviour from partially fair (0.43, 95%CI: 0.32, 0.55; ES = 0.19) and unfair (0.31, 95%CI: 0.25, 0.55; ES = 0.26) dictators than fair dictators. The mean log likelihood value across all trials and participants was -2.66 and median was -2.45 (range -8.32 - -0.02).

Following analyses of model fit, we wanted to assess the external validity of the model by checking that it was able to reproduce our behavioural results [19].

## Behavioural results for real data

A full analysis for the real data can be found in a previous paper [19]. In sum, Barnby et al., found that pre-existing paranoia increases harmful intent attributions (0.35, 95%CI: 0.15, 0.54), as does increasingly unfair dictator behaviour (2.00, 95%CI: 1.82, 2.18), and initially unfair dictator exposure leads to lower overall harmful intent attributions (-1.17, 95%CI: -1.52, -0.83). For self-interest, only increasingly unfair dictator behaviour (4.59, 95%CI: 4.26, 4.93), and initial dictator exposure (-0.71, 95%CI: -1.02, -0.39) were associated with attributions. These results should be the baseline comparison to assess our model's ability to generate simulated data in section 2.3.

## Behavioural results for simulated data

Simulated participants' (n = 1754) harmful intent attributions and self-interest attributions were only slightly negatively correlated with each other overall (rho = -0.06, p < 0.001). Simulated attributions were also highly correlated (harmful intent, rho = 0.83–0.9; self-interest, rho = 0.64–0.7, ps < 0.0001) with real participant attributions for all levels of GPTS score (as defined in [19]).

Along the continuum, pre-existing paranoia in all dictator conditions increased harmful intent attributions (Unfair: 0.22, 95%CI: 0.18, 0.25; Partially Fair: 0.19, 95%CI: 0.16, 0.23; Fair: 0.18, 95%CI: 0.15, 0.22). Along the continuum, pre-existing paranoia was only associated with slightly reduced self-interest attributions in unfair dictators only (-0.08, 95%CI: -0.12, -0.04), using individual cumulative link models with age and first dictator exposure as additional variables. A global model (Intercept: 0.01, 95%CI: -0.04, 0.05) reiterated the primary findings from the real behavioural data, namely, that paranoia increases harmful intent (0.11, 95%CI: 0.07, 0.15), as does increasing unfairness of Dictator (0.33, 95%CI: 0.33, 0.34) and being exposed to an initially unfair dictator (-0.34, 95%CI: -0.42, -0.27). Conversely, pre-existing paranoia did not affect self-interest attributions, but self-interest attributions were affected by unfairness of dictator (0.74, 95%CI: 0.73, 0.75) and simulated participants being exposed to a more unfair dictator first (-0.19, 95%CI: -0.25, -0.13). Fig 2 visually describes behavioural results for simulated data.

As the model was both found to be valid regarding it's predictive and generative performance [24], we then proceeded to assess whether latent parameters of uncertainty and learning rate varied by live attributions and pre-existing paranoid beliefs.

## Latent parameters associated with social inferences and pre-existing paranoia

Initial uncertainty over initial harmful intent (uHI0) was associated with age (-0.002, 95%CI: -0.002, -0.001) and pre-existing paranoia (0.04, 95%CI: 0.02, 0.05) but not sex. Likewise, initial uncertainty over initial self-interest (uSI0) was also associated with age (-0.002, 95%CI: -0.002, -0.001) and pre-existing paranoia (0.03; 95%CI: 0.01, 0.04), but not sex (Fig 3C). Uncertainty of partner policies (uΠ) was associated with pre-existing paranoia (0.09, 95%CI: 0.08, 0.10) but not age nor sex. Finally, participant's learning rates (η) were associated with age (0.003, 95% CI: 0.003, 0.004; ES = 0.05) but not pre-existing paranoia nor sex. See Fig 3 for visualisation of

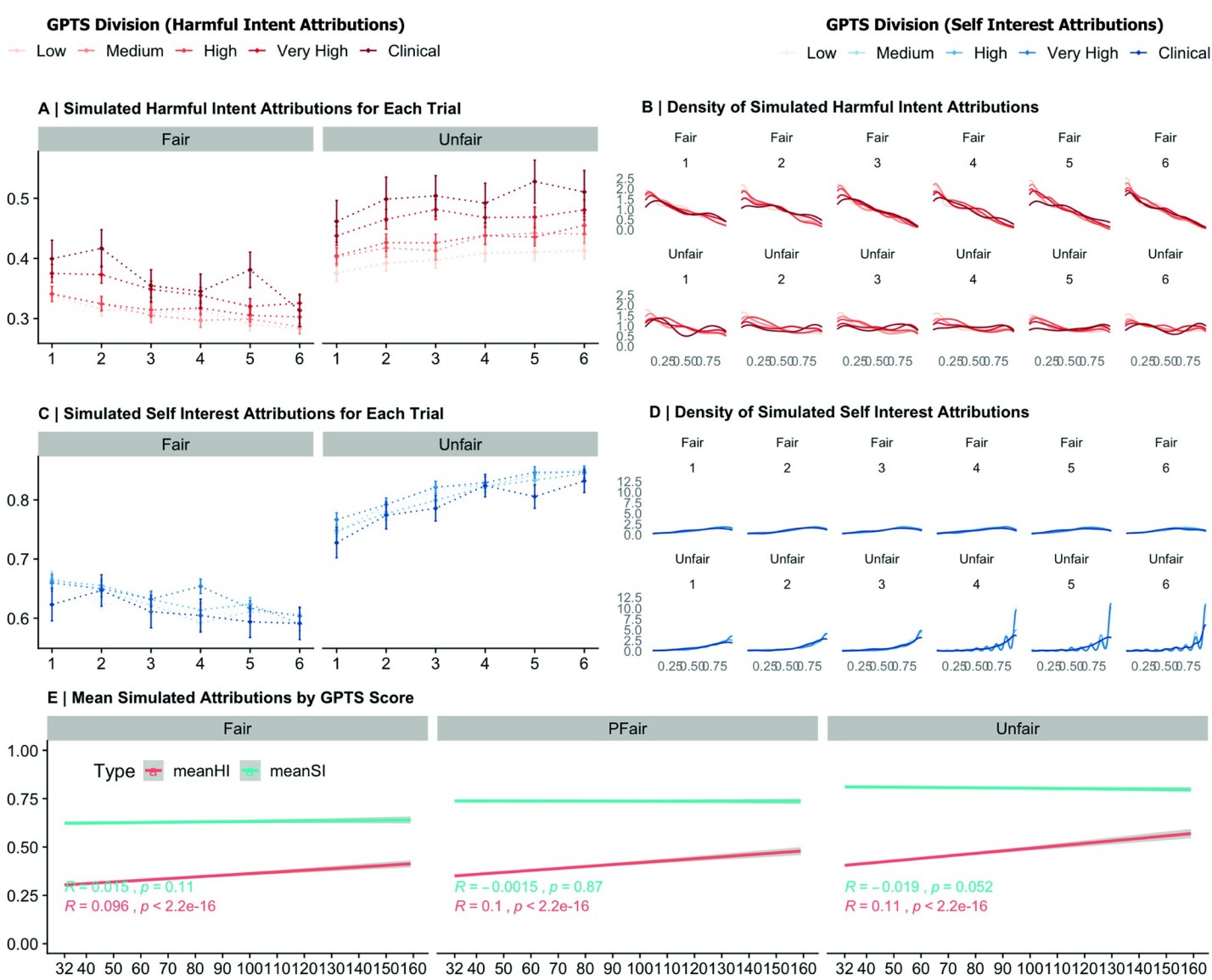

**Fig 2. Simulated Behavioural Data. (A)** Generated Harmful Intent (HI) attributions for simulated participants at each level of paranoia at each trial within fair and unfair dictators. Dots represent the mean for each level of paranoia. Lines represent the 95% confidence interval. **(B)** Generated density distributions for simulated participant HI attributions (red) for each trial (1–6) within unfair and fair dictators for each level of paranoia. **(C)** Generated Self-Interest (SI) attributions for simulated participants at each level of paranoia at each trial within fair and unfair dictators. Dots represent the mean for each level of paranoia. Lines represent the 95% confidence interval. **(D)** Generated density distributions for simulated participant SI attributions (blue) for each trial (1–6) within unfair and fair dictators for each level of paranoia. **(E)** Smoothed linear splines for both simulated participant harmful intent and self-interest attributions by prior paranoia (minimum score = 32).

initial and overall uncertainties and learning rates over pre-existing paranoia and in-the-moment attributions.

When assessing predictors of simulated attributions we found that simulated harmful intent attributions were associated with greater uHI0 (0.35, 95%CI: 0.31, 0.39), lower learning rates (-0.20; 95%CI: -0.34, -0.07), older age (0.004, 95%CI: 0.002, 0.006), greater pre-existing paranoia (0.09, 95%CI: 0.05, 0.13) and greater partner policy uncertainty (0.12, 95%CI: 0.08, 0.16). Mean simulated self-interest attributions were associated with lower uSI0 (-0.29, 95%CI: -0.31, -0.27), younger age (-0.003, 95%CI: -0.005, -0.001; ES = -0.05), and lower partner policy

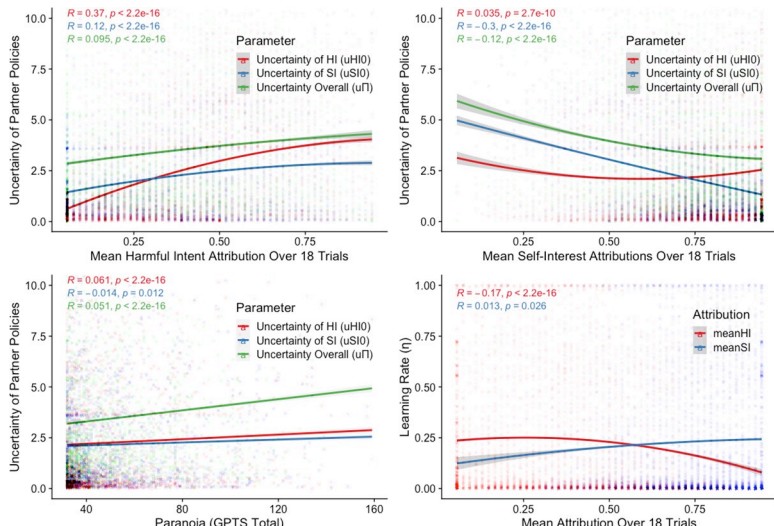

**Fig 3. Spearman rank correlations between uHI0, uSI0, uΠ, and η, and pre-existing paranoia and in-the-moment attributions. (A)** Quadratic fit for uncertainty of partner policies across the mean harmful intent attributions scored over 18 trials. **(B)** Quadratic fit for uncertainty of partner policies across the mean self-interest attributions scored over 18 trials. **(C)** Linear fit for uncertainty of partner policies across GPTS scores. **(D)** Quadratic fit of learning rate by mean attributions scored over 18 trials.

uncertainty (-0.13, 95%CI: -0.16, -0.10) but there was no effect of pre-existing paranoia, learning rate or sex.

In addition, we examined parameters in relation to the GPTS subscales, specifically its two main components, Social Reference and Persecutory Ideation, and when dividing the GPTS into scales of Conviction, Preoccupation, and Distress. When running spearman correlations, all subscales were highly correlated with each other (rho = 0.76–0.97, ps < 0.001), and all subscales were associated with the parameters to the same magnitude and direction as the GPTS total score (S7 Fig & S8 Fig).

There were also strong negative correlations between harm-intent, selfishness and policy uncertainties on the one hand, and model-fit on the other (k = -0.52, -0.43, -0.39 respectively, all p ~ 0.0).

## Network modelling

Finally, we explored the relationships between latent parameters generated by the model and pre-existing paranoid beliefs (Fig 4A), as well as the changes to network structure between parameters when moderated over pre-existing paranoia (Fig 4B), to assess whether pre-existing paranoia altered specific parameter relationships when controlling for all other relationships. We observed a strong positive partial correlation between the probability of attributing harmful intent (pHI0) and the probability of attributing self-interest (pSI0) (although see below), as well as a strong negative partial correlation between pHI0 and the uncertainty of harmful intent attributions (uHI0). Conversely, there was a strong negative partial correlation between pSI0 and the uncertainty of self-interest attributions (uSI0). Pre-existing paranoia was only positively partially correlated with pHI0 and uncertainty over partner policies (uΠ).

When moderating over pre-existing paranoid beliefs, we found that the edges between pSI0 and learning rates (η), and pHI0 and uΠ became more positively associated, and the relationship between pHI0 and η became more negatively associated when pre-existing paranoia increased (Fig 5).

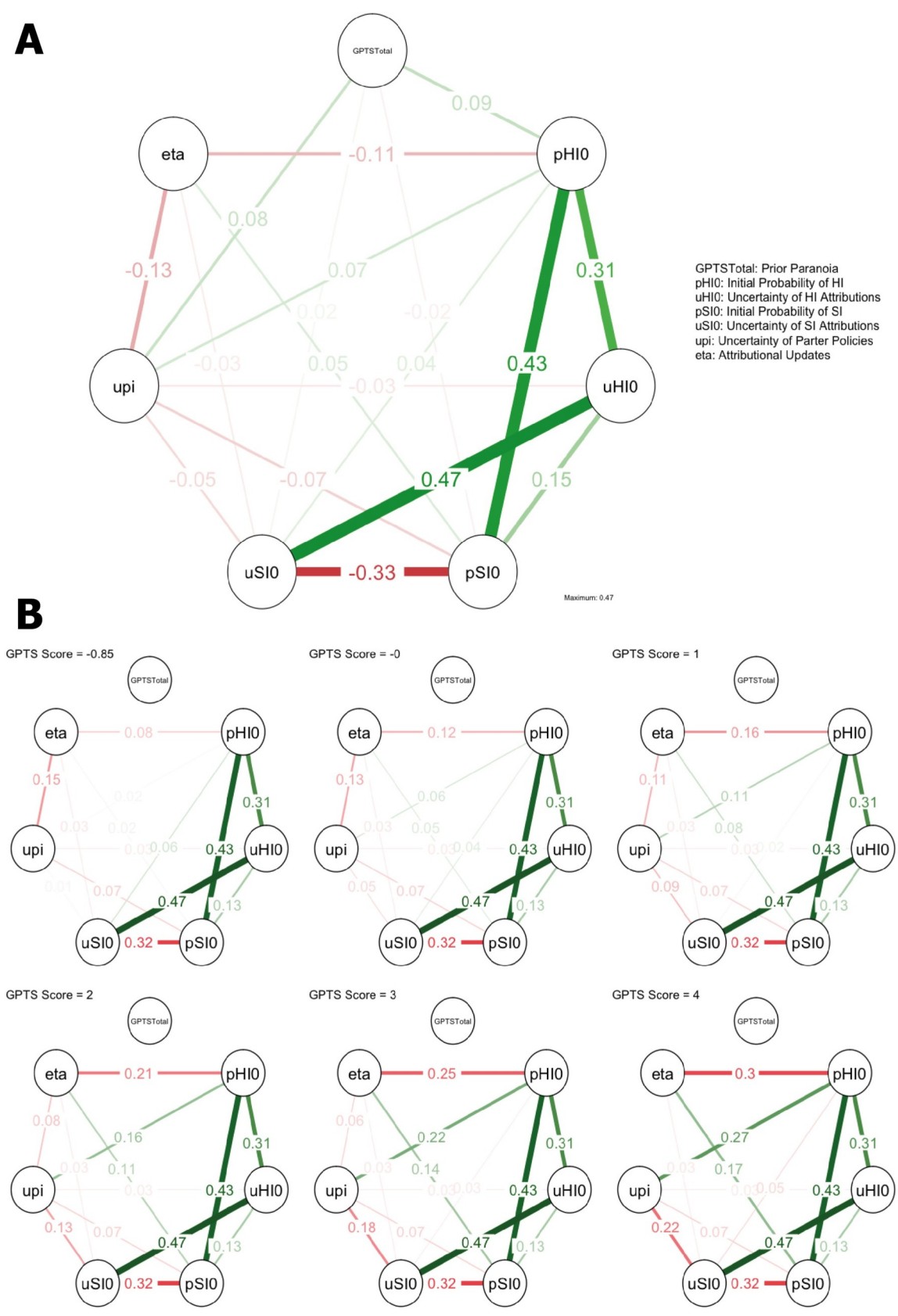

**Fig 4. Mixed Graphical Models. (A)** Gaussian Graphical Model of latent parameters and prior paranoid beliefs. **(B)** Moderated Network Model between latent parameters when moderated over prior paranoia from low to high Z-scores (-0.85–4). Red edges = negative association; green edges = positive association.

uHI0, pHI0 and pSI0 had most influence in the overall network, and all edges were found to be accurate following bootstrapping (S2 Fig).

Due to the non-normal data distribution, and that model-fitting may itself induce correlations between estimated parameters in the population, we ran our model on simulated data, generated using parameters that captured a substantial range of each major cluster in our real data (low, medium & high cluster; see methods). We then ran the same network modelling procedures as the main analysis. Edges recovered in the networks based on simulated data that were in the same direction as our 'true' data indicated that relationships in the 'true data' may be artefactual. However, the only relationships recovered in the same direction were i) a positive correlation between pHI0 and pSI0 in our 'medium cluster' simulated network and ii) an inconsistent negative correlation between pSI0 and uSI0 in our 'high cluster' network (S3 Fig). Therefore, these specific relationships should be treated with some caution. Full details of the procedure are listed in Methods.

In addition, it may be that the complexity of the network itself is generating moderated effects between parameters due to redundant inclusion of variables around the relationships of interest [25]. To control for this possibility, we ran the Moderated Network Model with two, three, four, and five variables in the network to check whether the relationships between learning rate (eta), partner policy uncertainty (upi), and probability of attributing harmful intent (pHI0) still existed in absence of other relationships (S4 Fig). We found that the variables in question were retained and were moderated in the same direction by GPTS score in all network models.

## Discussion

We wanted to test whether differences in uncertainty and learning in those with high paranoia observed in prior non-social models also apply to social learning. We were also interested in whether threat and non-threat social attributions would show different relationships to uncertainty and learning parameters and potential interactions with levels of pre-existing paranoia. To achieve this, we fitted a computational model to a multi-round Dictator task, which allowed us to estimate the uncertainty of partner choices, or policy, and the learning rate over social encounters made by participants, using large-scale behavioural data previously reported [19]. Our model was able to reproduce the behavioural effects previously reported, in addition to showing adequate fit across trials, divisions of GPTS scores, and the different types of Dictator that each participant encountered in the task. Our results finesse this finding, in that harmful intent and self-interest attributions are differentiated by their relationship to uncertainty in the model and moderated to different degrees of pre-existing paranoia.

Overall, we found that paranoia was associated with increased uncertainty regarding another's policy (u$\Pi$), and baseline uncertainty parameters over harmful intent (uHI0) and selfishness intent (uSI0) to a lesser extent. In other words, pre-existing paranoia led to weaker, more variable initial assumptions about partners, and continued attributions of harmful intent regardless of the behaviour observed. Attributing higher harmful intent regardless of pre-existing paranoia was associated with higher u$\Pi$ and uHI0, and lower learning rates ($\eta$). Conversely, attributing higher self-interest was associated with reduced u$\Pi$ and uSI0, and greater $\eta$. In other words, self-interest attributions were made with more precision and were more informed by previous experience than harmful intent attributions, which were more uncertain

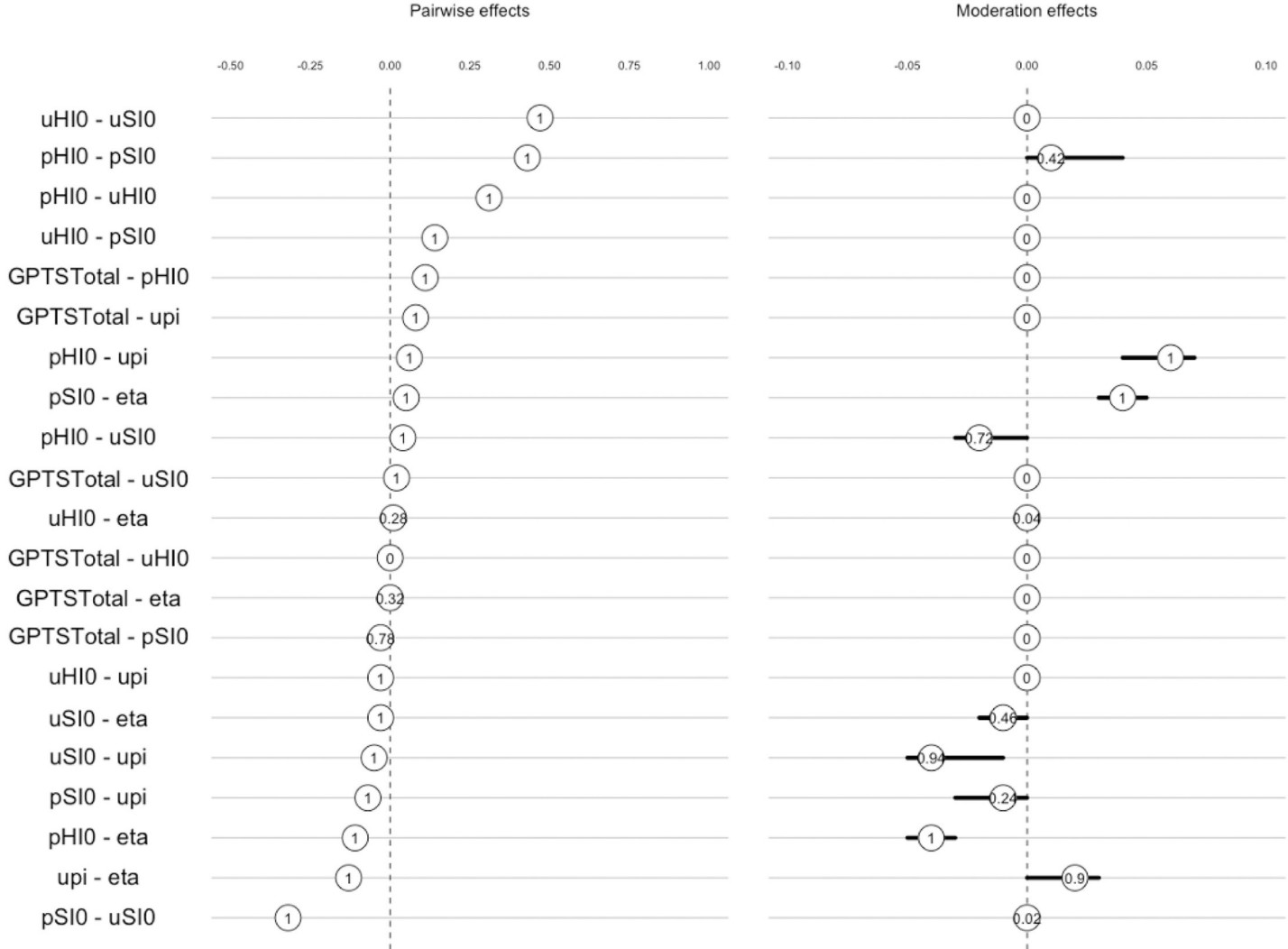

**Fig 5. Moderation effects of pre-existing paranoia on edges within the Moderated Network Model (Fig 4B).** The left panel displays the pairwise effects–the overall relationship between the parameters in the Gaussian Graphical Model of parameters—and the right panel shows the moderation effect of GPTS score on the pairwise effects–the influence of variable GPTS scores on the relationship between parameters. Both are shown with 95% confidence intervals of the bootstrapped sampling distributions. The number at the centre of the sampling distribution is the proportion of bootstrap samples in which a parameter has been estimated to be nonzero [26].

and based more on social context. Interestingly, in convergence with prior studies [14], we found that baseline GPTS score was negatively correlated with model fit, suggesting pre-paranoid beliefs lead to less 'Bayesian ideal' behaviour in a social task.

Network modelling suggested that pre-existing paranoia most notably moderated the relationship between pHI0 and η, and pHI0 and uΠ. Across the board, regardless of pre-existing paranoia, uHI0 and pHI0 were positively partially correlated, whereas pSI0 and uSI0 were negatively partially correlated. High probabilities of harmful intent attributions were associated with reduced precision, but high probabilities of self-interest attributions were associated with increased precision.

Our findings converge with the idea that pre-existing paranoia (a higher-level belief about others) influences momentary inferences regardless of a partner's behaviour [20]. Prior non-social evidence [27] and simulation studies [28] conclude that delusional beliefs can be in part explained by a deficit in holding stable beliefs about the world. Models developed from social

simulations have suggested that those in high risk states may show more uncertainty, whereas full-fledged delusions may lead to rigid, highly certainty beliefs about others [21].

Additionally, our data converge with Reed and colleagues [13] to suggest that pre-existing paranoia makes individuals more sensitive and more reliant on current social environmental conditions. Relevant evidence using non-social tasks has been mixed: some have suggested that pre-existing paranoid beliefs lead to reduced beliefs updates when playing a non-social associative task [14], whereas others suggest paranoia leads to rapid associative updates as if participants expect continually changing rules [13].

Finally, we suggest attributing higher harmful intent to social actions may engender an adaptive cognitive state to remain vigilant about a new partner's future harmful intentions. Previous experimental work using a predictive task to assess whether individuals could infer their partners future choices, and their impressions of their partner, found beliefs about 'bad' actors were more volatile to allow for correcting an impression [29]. Our findings may extend these conclusions. Indeed, better encoding of harmful agents may be evolutionarily adaptive within-individuals [30]. In contrast, selfish impressions do not pose the same need for vigilance.

We might argue here that if paranoia could be explained solely by domain-general effects (as argued by [13]), we would have expected pre-existing paranoid beliefs to moderate the relationship between learning, uncertainty and both social attributions in the same direction, or find no difference between the two social attributions. The role of delineated social mechanisms in human and animal cognition in shaping interpersonal interaction [31] during evolutionary and individual development warrants a clearer focus on how alterations to these social mechanisms alone or in tandem with general cognitive functions may shape psychiatric disorder.

The parameters identified in our study may be useful probes to modulate in future experiments. Psychopharmacological experimental work [32] and PET observations [33] suggest dopamine may be crucial in the transmission of social threat and attribution of harmful intent, and D2/3 receptor transmission may modulate the relationship of policy precision [34]. We hypothesise that interventions to induce heightened dopamine transmission in midbrain regions specifically at D2 receptors (e.g. from drug use) will result in heightened probabilities of harmful intent attributions, reduced precision over harmful intent attributions, reduced learning rates about harmful intent, and greater uncertainty of partner policy overall, although conversely we may expect greater precision, greater learning, and reduced uncertainty over non-threat related attributions and behavior, such as self-interest attributions.

We should note some limitations. While we partnered participants against genuine people, we cannot capture all social nuance that might be present in a real-world interaction. There may be a host of other social factors that may influence learning rates and uncertainty over inferred social intentions in paranoia such as eye contact and multiple social actors [35], and we feel this is an important topic for future research. Additionally, while not a substantial limitation for us to use Bayesian inference rather than its common approximations (e.g. constant learning rates that differ for the two attributes) given the small amount of data per dictator, using this data we were not able to fit separately the participants' own noisy reporting (emission noise) vs. noisy aspects of their generative model of others (esp. dictator policy uncertainty). Future work may consider validating both inferential and response models using a greater amount of data per participant and specially designed conditions.

## Conclusion

In sum, we suggest that pre-existing paranoia may differentially moderate parameters that govern harmful-intent attributions compared to those that moderate self-interest attributions. We

modelled large-scale behavioural data using a 'Bayesian-brain' model of others intent and found that the model fitted adequately across the range of pre-existing paranoia was able to replicate prior behavioural data. Assessing latent parameters suggested that pre-existing paranoia may be dissociated from a potentially more universal mechanism between the magnitude and uncertainty of harmful intent attributions. Pre-existing paranoia led to higher attribution of harmful intent for any new social interaction, but it also rendered behaviours less dependent upon inferences about previous encounters. Importantly, we show that pre-existing paranoia should not be assumed to apply to all inferential processes equally. Parameters derived from our models may be useful in describing individual variation in psychobiological mechanisms in future experiments.

## Methods

### Ethics statement

The original studies were approved by the Kings College London ethics board (Study 1: MRS-17/18-8312; Study 2: LRS-18/19-9281) and preregistered (Study 1: http://aspredicted.org/blind.php?x=8cj8zk; Study 2: http://aspredicted.org/blind.php?x=ub9z2x). Full formal consent was obtained by participants through our online platform.

All data in Study 1 were collected in September 2018 and for Study 2 in February 2019 using Prolific Academic (hereafter Prolific; www.prolific.ac), an online crowdsourcing platform. Data from both studies were combined to perform the computational analysis. 1998 participants were recruited in total from both studies, but only 1784 participants were used in analysis due to drop out or failing control questions between the baseline administration of the GPTS and the serial Dictator game.

Prior to taking part in both studies, participants were informed that their participation was voluntary and were required to tick a box that consented to the authors using their anonymous data for research purposes. Using Prolific allowed us to rapidly recruit a more demographically diverse sample of participants than recruitment from our social media or university networks [36]. We included participants from the UK who were fluent in English and had no current or history of mental illness.

Participants first completed the Green Paranoid Thoughts Scale (GPTS; [23]). Participants were asked to indicate the extent of feelings described in 32 statements using a Likert Scale of 1 to 5, where 1 = Not at All and 5 = Totally. Scores can range from 32–160, with higher scores indicating a greater degree of paranoia. The GPTS was chosen as a suitable measure as it includes both core aspects of the definition of paranoia [1]: social concerns about others and perception of intended harm. It has also shown to be the most reliable and valid scale for measuring paranoia across the clinical and non-clinical spectrum [37]. Total paranoia scores were obtained for each participant by summing the response scores to all questions, comprising both the social reference and the persecution scales. Hereafter, this variable is referred to as 'paranoia'.

After completing the survey, and in keeping Raihani and Bell [16,17] we allowed a minimum interval of 7 days to elapse before inviting participants to take part in the Helsinki Summit.

We developed a within-subjects, multi-trial modification on the Dictator game design used in previous studies to assess paranoia [19]. Each participant played six trials against three different types of dictator. In each trial, participants were told that they have been endowed with a total of £0.10 and their partner (the dictator) had the choice to take half (£0.05) or all (£0.10) the money from the participant. Dictators were set to either always take half of the money, have a 50:50 chance to take half or all of the money, or always take all of the money, labelled as Fair, Partially Fair, and Unfair, respectively. The order that participants were matched with dictators was randomised. Each dictator had a corresponding cartoon avatar with a neutral

expression to support the perception that each of the six trials was with the same partner. After each trial, participants were asked to rate on a scale of 1–100 (initialised at 50) to what degree they believed that the dictator was motivated a) by a desire to earn more (self-Interest) and b) by a desire to reduce their bonus in the trial (harmful intent). Following each block of 6 trials participants were asked to rate the character of the dictator overall by scoring intention again on both scales. Therefore, participants judged their perceived intention of the dictator on both a trial-by-trial and partner level.

After making all 42 attributions (two trial attributions for each of the 6 trials over 3 partners, plus three additional overall attributions for each partner), participants were put in the role of the dictator for 6 trials–whether to make a fair or unfair split of £0.10. Participants were first asked to choose an avatar from nine different cartoon faces before deciding on their 6 different splits. These dictator decisions were not used for analysis but were collected in the first phase of the game to match subsequent participants with decisions from real partners using an 'ex-post' matching design.

The modification to the original dictator game design allowed us to track how partner behaviour, order of partner, and whether attributions were stochastic or consistent as pre-existing paranoia changed. All participants were paid for their completion of the GPTS, regardless of follow up. Participants were paid a baseline payment for their completion (See S1 Text for the task instructions given to participants and task schematic)

## Analysis

**Computational model.** Our computational analysis was not preregistered. All data and analysis scripts are available on the Open Science Framework (https://osf.io/24urf/).

We considered a belief-based modelling framework. Beliefs here are not necessarily, representational propositions but 'effective' beliefs, that is, engrams encoding probability distributions, necessary for performing approximate probabilistic information processing. How such engrams may be implemented in the brain is a matter of debate. Much evidence suggests that the brain encodes expectations about what will be observed [38,39] but it is less certain as to how the shape of belief distributions is represented. Some hold that the width of belief distributions is represented as activity explicitly reflecting precision, while others claim that the brain may estimate uncertainty 'on the fly' by sampling from available memories or other representations [40,41]. In our case, the nature of these engrams of expectation and uncertainty is irrelevant, but their function to provide weights for the updating of different beliefs is crucial. Social actions in the Dictator game such as 'my partner decided to give me nothing' can be used to infer the probability or degree of the partner's 'hostility' or self-interest. We further assumed that propositional cognition, such as ratings on a scale, reflect noisy sampling of the neurally encoded effective beliefs.

We model effective beliefs about dictator's attributes as ranging along two dimensions, harmful intent and self-interest attributions. We can discretise them into Likert-like bins, as long as the bin resolution is sufficient. Here, we discretised along 9 bins, from 'totally altruistic' ($HI = 1$, $SI = 1$) to 'totally antisocial' ($HI = 9$, $SI = 9$). The prior beliefs about Others formed the most important part of our modelling, parametrized by a central tendency parameter $HI_0$, $SI_0$ and an uncertainty $uHI$, $uSI$ along each dimension. Inference over such discrete distributions can be conveniently parametrized the Binomial distribution with $n$ bins and parameter $p$, sharpened (or blunted) by an uncertainty parameter $u$:

$$Pk \propto Bin(k; p, n)^u$$
$$:= NB(k; p, u, n)$$

(1)

When the exponent in Eq 1 is greater than 1, the distribution keeps the same mode but is sharpened; when less than 1, it is blunted. The prior belief over both *HI* and *SI* can then be written as a product of the independent prior probabilities, $p^{(0)}_{HI} * p^{(0)}_{SI}$. This assumption of independence is conservative, minimizing the number of free parameters.

In order to make inferences based on the feedback they get from dictators, participants must also hold a correspondence between attributes and behaviours. We emphasise that participants hold maps *from attributes to behaviour*, and not directly from observations of returns to attributes. Thus, they have to invert these maps to update their beliefs, which will typically result in asymmetric belief update depending on further detail (so that Eq. uses full joint probabilities, breaking the initial independence $p(0)(HI,SI) = p(0)HI * p(0)SI$. To build a map from attributes to behaviour that could capture a full range of possibilities, and thus could be used to describe all participants, we first provided for a range of possible dictator behaviours, discretizing returns using a similar resolution as attributes. *HI* = 0, *SI* = 0 corresponded to the attributions of each participant which would result in a dictator preference to give *r (return)* = *n* (of *n*, i.e. 100%) to the participant ('self-sacrifice'). At the top end of the scale, *HI* = 1, *SI* = 1 were attributions resulting in a high preference for giving *r* = 1 (of *n*, i.e. 0%) to the participant. Very high *SI* or *HI* with moderate scores on the other dimension were sufficient for a substantial probability of *r* = 1 of n (i.e. 0%). We implemented this general template map $\pi_{gen}$ using *a priori* fixed parameters. This is illustrated in Fig 6, and the corresponding equations (Eq 2) are

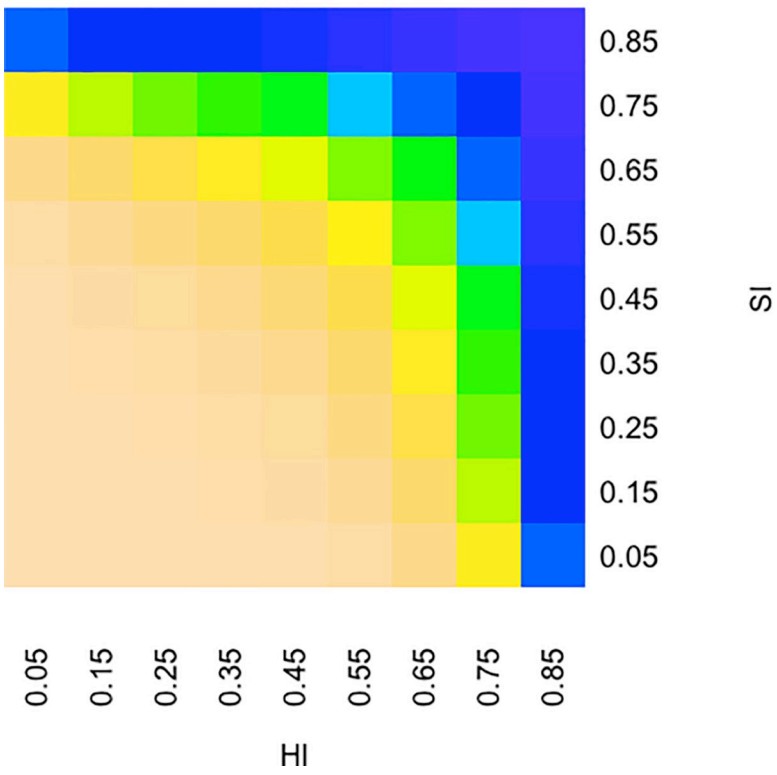

**Fig 6. Mean partner policy depending on attributes.** Okra: preference for returning a large amount to the participant. Blue / purple: preference for returning very little.

given below for completeness.

$$\pi_{gen}(r; HI, SI) = \pi_{gen}(r; HI)\pi_{gen}(r; SI)$$

with:

$$\pi_{gen}(r; HI) = NB(r; 1 - p_{init} - \delta p\ HI, u_{init} - 2\delta p(HI + 1), n_\pi)$$
$$\pi_{gen}(r; SI) = NB(r; 1 - p_{init} - \delta p\ SI, u_{init} - 2\delta p(SI + 1), n_\pi)$$
$$\delta p = \frac{1 - 2p_{init}}{n_\pi - 1}$$
$$p_{init} = 0.05$$
$$u_{init} = 2.5$$
$$n_\pi = 9$$

(2)

However, these preference probabilities were not all available to the dictator. This means that every modelled participant had the same basic repertoire of attribute-behaviour available to them, and that *HI* and *SI* can be seen as ideographically scaled (each attribute can be scaled on an individual basis). Then, one additional parameter was introduced, to quantify individual variation in the consistency agents expected between attitudes and behaviours. On the basis of previous work, a small, fixed lapse rate $\xi = 0.02/n^2$ was also added to increase numerical stability. This was another noise or uncertainty parameter $u_\pi$, over the dictator's policies. We thus used:

$$\pi(r; SI, HI, u_\pi) \propto \pi_{gen}(r; SI, HI)^{\frac{1}{u_\pi}} + \xi$$

(3)

This completes the participants' generative beliefs of the Dictator's behaviour, and provides for exact, numerically tractable Bayesian updates in the beliefs of the participant when they receive feedback. For each potential attribute pair (*HI*, *SI*) of the Dictator:

$$p_t(HI, SI) = \frac{\pi(r; HI, SI)p_{t-1}(HI, SI)}{\sum_{HI', SI'}\pi(r; HI', SI')p_{t-1}(HI', SI')}$$

(4)

We then considered that participants inform their beliefs about the second dictator they see by what they learnt about the first one, and so on. The simplest approximation is to add a small admixture of the posterior beliefs about the last Dictator to the priors they used for this last dictator, weighing this posterior by an individually fitted learning rate η.

Finally, as mentioned above, reported attributions were taken to be sampled from the underlying belief distributions. We note that in our experiment it is not possible to clearly distinguish between uncertainty participants display due to their own noisy cognition, as opposed to noisy decision-making that they expect their partners to display. In our case, both of these would result in greater participant uncertainty and noisier reporting of inferred attributes.

Overall, therefore, each participant is characterized by six parameters: the central tendencies of their initial priors about Harm and Selfishness intent, HI0, SI0, their corresponding uncertainties, uHI, uSI, their belief about (in)consistency of the Dictators' actions, uΠ, and their learning rate about Dictators, η.

The models were fitted with Maximum A Posteriori (MAP) estimation, i.e. penalizing maximum likelihood with a weak, regularizing prior restricting parameter values to their psychologically meaningful ranges (e.g. learning rate between 0 and 1, etc.). A grid-search approach on parameter values was combined was followed by gradient-ascent on MAP to minimize the chance of missing important MAP maxima.

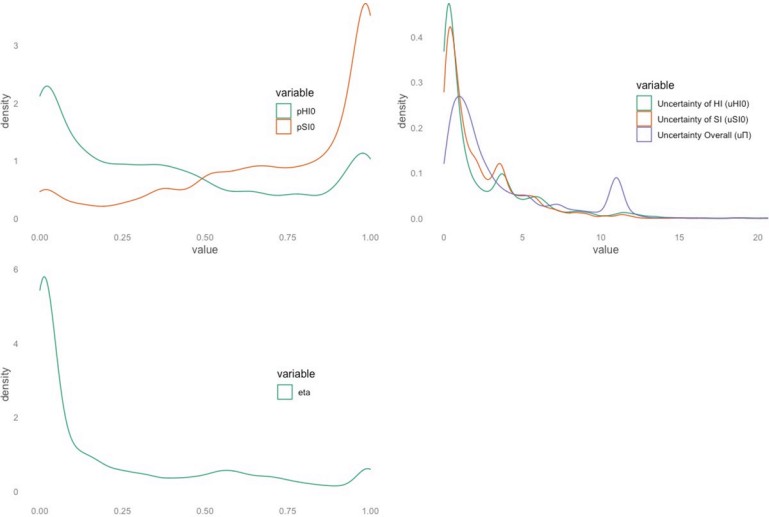

**Fig 7. Smoothed density distributions of the fitted parameters derived from the computational model.**

**Information theoretic analysis.** We primarily aimed to assess our candidate model's predictive and generative performance. The ability of a model to simulate data is necessary to assess its validity and falsification [24]. This centred around our ability to replicate our effects documented from our previously reported behavioural results [19]. We then aimed to assess our model fitting by using the log-likelihood values across trials, dictators, and divisions of GPTS score (z scaled, continuous GPTS scores). Following this, we aimed to statistically interrogate the generated data in the same manner as we did with the behavioural data. Density distributions of the fitted parameters can be found in Fig 7.

We used an information-theoretic approach for all analyses unless otherwise stated. All analyses were performed in R (version 4.0.0; 41) on an Apple OSX operating system (Mojave, 10.14.6). We used general linear, mixed linear or cumulative link models for continuous or ordinal outcome variables, respectively. We analysed each model using multi-model selection with model averaging (described in [19]). The Akaike information criterion, corrected for small sample sizes (AICc), was used to evaluate models, with lower AICc values indicating a better fit [42]. The best models are those with the lowest AICc value. To adjust for the intrinsic uncertainty over which model is the true 'best' model, we averaged over the models in the top model set to generate model-averaged effect sizes and confidence intervals [43]. In addition, parameter estimates, and confidence intervals are provided with the full global model to robustly report a variable's effect in a model [44]. This used package [45]. All visualisations were generated using the package 'ggplot2' [46].

Specifically, we used cumulative link models to calculate the behavioural effects in the simulated data for individual dictators. All variables of 'Prior paranoia' listed are scaled and centred GPTS total z-scores.

$$\text{Ordinal HI/SI} \sim \text{Prior paranoia} + \text{age} + \text{order}$$

In our overall global behavioural model, our mixed cumulative link model didn't converge as the Hessian was not a positive definite, and thus we used a linear mixed model (package "lme4"; [47]) with multimodal averaging instead:

$$\text{Mean HI/SI over 18 trials} \sim \text{Prior paranoia} + \text{age} + \text{dictator} + \text{order} + (1|\text{ID})$$

To assess our latent uncertainty parameter associations (uHI0/uSI0/uΠ), and attributional updates (η), we used individual general linear models for each parameter with multimodal averaging, with their respective relevant simulated attributions:

$$\text{uHI0/uSI0} \sim \text{Prior paranoia} + \text{age} + \text{sex}$$

$$\text{uΠ} \sim \text{Prior paranoia} + \text{age} + \text{sex}$$

$$\text{η} \sim \text{Prior paranoia} + \text{age} + \text{sex}$$

Global models to assess predictors of simulated harmful intent and self-interest attributions, linear mixed models with participant identification number (ID) as a random effect were used with multimodal averaging:

$$\text{Mean simulated HI/SI} \sim \text{Prior paranoia} + \text{age} + \text{sex} + \text{uHI0/uSI0} + \text{η} + \text{uΠ} + (1|\text{ID})$$

In our models, all continuous scale scores were centred and scaled to produce Z values. All model statistics reported are beta coefficients unless otherwise stated.

**Network modelling.**    We performed network modelling of the derived parameters using Gaussian Graphical Models and Moderated Network Modelling (package "mgm", version 1.2–8; [26]). Graphical models allow insight into the relational and dependent patterns in multivariate data, specifically as they relate to pairwise relationships [26]. Fundamentally, graphical models employ partial correlations between variables inputted into the network and allow a 'model-free' approach to generate structure from the data itself.

We first estimated an overall network model that included 8 variables: all latent parameters and prior paranoia using bootnet (version 1.4.3; [48]. All variables were z-scaled. We used the 'ggmModselect' function in bootnet that allows a search for the optimal gaussian graphical model through minimisation of the extended Bayesian information criterion (EBIC) in a stepwise manner. This has been shown to lead to consistent estimates [49]. The function first obtains 100 regularised possible models using least graphical absolute shrinkage estimation (glasso) and then refits all models without regularisation. The best models are based on their EBIC values. All possible models are then tested stepwise by adding or removing edges and reassessing the model fit. When edges being added or removed do not change the EBIC, the algorithm stops.

We also estimated the accuracy of edges and stability of edges and centrality metrics in the overall network. Accuracy of the network is calculated in 'bootnet' by resampling each pairwise edge 1000 times with a resample-with-replacement method. The original sample is then compared with the bootstrapped sample. Small confidence intervals and similarity between the bootstrapped and original sample represents good accuracy of the network. Stability of the network is calculated using a 'case-dropping' bootstrap method in 'bootnet'. Centrality metrics are recalculated for the whole network after removing cases in the population 10% at a time, with stability of each centrality measure considered good if it does not drop below 0.5, and excellent if it does not drop below 0.7 [50]).

Finally, we assessed the changes to edge weights between nodes in the network when moderating over prior paranoia using Moderated Network Models (MNM). This allows for changes in edge weights to be estimated as a moderator–here prior paranoia–is set to different z-score values. This method allows the effect of the moderator to be estimate without splitting the data using a linear moderation effect (for full details of the moderation method see: [26]). We also ran bootstraps on the moderation effect estimates to assess the stability of the moderation effects.

The null hypothesis against which a network should be tested is not that the nodes are uncorrelated, but that the correlations are not accounted for by artefacts of the measuring process. The network does not represent the marginal distributions of the node variables in the population, but the relationships between them. Therefore, the appropriate null hypothesis is whether the measuring process would artificially induce the observed correlation structure if the node variables were sampled from uncorrelated regions of real distribution. This possibility is well recognised in cognitive modelling, as correlations may be induced by so-called 'trading off' of parameters against each other [51]. Insignificant correlations in the simulated model, or correlations in the opposite direction to those found in real data, suggest that the relationships actually found in the data are not due to model-fitting artefacts.

To test this, we considered the key regions of high density in the joint distribution of fitted parameters (S5 Fig). This produced high (pHI0 = 0.01–0.98; pSI0 = 0.8–0.98), medium (pHI0 = 0.25–0.5; pSI0 = 0.5–0.75), and low-density (pHI0 = 0.01–0.2; pSI0 = 0.01–0.2) clusters of pHI0 x pSI0, and single dense cluster for each of uHI0, uSI0, eta, and upi, independent of other parameters. We thus formed three high-density clusters in the entire parameter space. From these, we sampled three simulated behavioural datasets of 200 pseudo-participants that included either high, medium, or low pHI0 and pSI0 with each of the other four parameters. Each simulated behavioural dataset was then run through the same MAP procedure as before, to produce re-fitted parameters: a grid-search approach on parameter values was followed by gradient-ascent on MAP. Correlations between refitted parameters from each pseudo-data set were then analysed to assess their strength, direction, and significance versus the real data (S6 Fig).

## Supporting information

**S1 Text. Instructions given to participants at the start of the serial Dictator game.**
(DOCX)

**S1 Fig. Task schematic of the serial Dictator game.**
(DOCX)

**S2 Fig.** (A) Node centrality–strength, closeness, betweenness, and expected influence–of the overall model (Fig 3A). (B) Accuracy of the network when bootstrapped with 1000 sampling-with-replacement. Grey shaded area represented confidence intervals. (C) Stability of the strength, closeness, and betweenness of nodes in the network. (D) Node centrality by each model moderated at different degrees of prior paranoia.
(DOCX)

**S3 Fig. Networks are derived from each batch of simulated (n = 200 each) and real (n = 1754) data.** Low, Med, and High represent the clusters of pHI and pSI parameters used to simulate the data based on the differential clustering found in Fig 7. Each bootstrapped network analysis used 1000 bootstraps.
(DOCX)

**S4 Fig. This analysis demonstrates that the key moderation effects of interest observed in the main analysis between pHI0, eta, and upi is still present despite other parameters being removed from the network.**
(DOCX)

**S5 Fig. Each cluster of densities between pHI0 and pSI0 were used to construct three independent models that represented 'low', 'medium', and 'high' clusters.**
(DOCX)

**S6 Fig. All attributions were scaled and centred before generating correlations to keep both on a common range, as real attributions were scored from 0–100 and simulated attributions were generated between 0–1.**
(DOCX)

**S7 Fig. An 'X' over a coefficient value denotes that it is non-significant.** All other values are significant at least at the $p < 0.05$ level.
(DOCX)

**S8 Fig. An 'X' over a coefficient value denotes that it is non-significant.** All other values are significant at least at the $p < 0.05$ level.
(DOCX)

## Acknowledgments

We greatly thank Dr Jonas Haslbeck and Dr Olivia Guest for their discussion during the preparation of this manuscript.

## Author Contributions

**Conceptualization:** Joseph M. Barnby.

**Data curation:** Joseph M. Barnby.

**Formal analysis:** Joseph M. Barnby, Michael Moutoussis.

**Funding acquisition:** Joseph M. Barnby.

**Investigation:** Joseph M. Barnby, Michael Moutoussis.

**Methodology:** Joseph M. Barnby, Michael Moutoussis.

**Project administration:** Joseph M. Barnby, Michael Moutoussis.

**Resources:** Joseph M. Barnby, Michael Moutoussis.

**Software:** Joseph M. Barnby, Vaughan Bell, Michael Moutoussis.

**Supervision:** Vaughan Bell, Mitul A. Mehta, Michael Moutoussis.

**Validation:** Joseph M. Barnby.

**Visualization:** Joseph M. Barnby.

**Writing – original draft:** Joseph M. Barnby.

**Writing – review & editing:** Joseph M. Barnby, Vaughan Bell, Mitul A. Mehta, Michael Moutoussis.

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
