## [Decision Letter · Decision Letter 0]

17 Jun 2020

Dear Mr Barnby,

Thank you very much for submitting your manuscript "Reduction in social learning and policy uncertainty about intentional social threat underlies paranoia: evidence from modelling a modified serial dictator game." for consideration at PLOS Computational Biology.

As with all papers reviewed by the journal, your manuscript was reviewed by members of the editorial board and by several independent reviewers. In light of the reviews (below this email), we would like to invite the resubmission of a significantly-revised version that takes into account the reviewers' comments.

I have an additional comment, regarding the estimation of network interactions. You can have mediated or directed interactions, and for this you normally use partial correlations. On the other hand you can have associations which go beyond pairwise interactions (regardless of the fact that they are mediated or not). This is the case when you have latent variables which jointly influence a subset of variables in the system, but you can also have higher order effects within the variables in the system, for example two variables sharing joint information on a third one. These types of interactions confound the estimation of pairwise interactions, also when conditioning for the action of other single variables. See for example

Hoffmann, T., Peel, L., Lambiotte, R., & Jones, N. S. (2020). Community detection in networks without observing edges. Science Advances, 6(4), eaav1478. doi:10.1126/sciadv.aav1478

Stramaglia, S., Cortes, J. M., & Marinazzo, D. (2014). Synergy and redundancy in the Granger causal analysis of dynamical networks. *New Journal of Physics*, *16*(10), 105003. http://dx.doi.org/10.1088/1367-2630/16/10/105003

Do you control for this type of interactions? Do you expect them?

We cannot make any decision about publication until we have seen the revised manuscript and your response to the reviewers' comments. Your revised manuscript is also likely to be sent to reviewers for further evaluation.

Sincerely,

Daniele Marinazzo

Deputy Editor

PLOS Computational Biology

Daniele Marinazzo

Deputy Editor

PLOS Computational Biology

Reviewer's Responses to Questions

**Comments to the Authors:**

Reviewer #1: The authors of “Reduction in social learning and policy uncertainty about intentional social threat underlies paranoia: evidence from modelling a modified serial dictator game” present a computational model to explain how pre-existing paranoia impacts social learning and perception of others’ behaviour.

This is a well written paper and tackles an important aspect of paranoia, namely one’s uncertainty about the others’ action policies. I appreciate the inclusion of in-silico data to reproduce the major behavioural trends observed empirically. I have a few questions related to the study design and the predictive validity of the model parameters.

1. Paranoia using the Green Paranoid Thoughts Scale (GPTS): First, different aspects of pre-existing paranoia might determine orthogonal aspects of social learning and decision-making (hence the question below related to subgroups). For example, an increased frequency in paranoid thoughts may relate to more uncertain beliefs about others’ intentions while increased conviction of paranoid thoughts may be predicted by overly high precision of predictions and therefore reduced learning rates and belief flexibility. Thus, my question is, does the GPTS scale you have used differentiate between these aspects of paranoia? And if so, what aspect of paranoia is predicted by which parameter of your model?

Second, was the assessment of paranoia performed at multiple time points? In some cases, a brief expression of paranoia may reflect a period of high stress and may be associated with an increased perception of volatility (and thus uncertainty about others) while a stable and prolonged expression of paranoia may reflect overly precise higher-level beliefs and reduced learning rates. Can you differentiate between state and trait?

2. Experimental Paradigm: It is not clear from the description of the task whether this was a reciprocal exchange or rather used a fixed input structure. Related to this, it unclear whether deception was used. It would help to include the task instructions.

3. Can the computational model parameters identify different mechanisms of paranoia affecting social inference and decision-making? The effect sizes related to the predictive validity of the model parameters seems to be rather low. Do the authors have an interpretation for this? Given the current effect sizes, do you think the model can in theory capture individual differences in what aspect of social learning is impacted by paranoia?

4. Interpretability of network modelling results: These are a very useful representation of the interactions between parameters but it would be helpful to filter out these interactions by which connections can be recovered through simulations. As you have pointed out, the interpretation of these results is confused by general correlations between the parameters of the model. These identifiability issues would be clarified using this simulations approach.

I also have a few minor questions and comments:

1. What do you make of the consistent age effects observed?

2. I had the feeling that precision and uncertainty are sometimes used interchangeably, but I think it is important to be precise about what aspects of learning they refer to. For example, one may be overly confident about expecting harmful intent while still being uncertain about the dictator strategy at a given moment in time (i.e., potentially due to an enhanced perception of volatility). These two things may also be mutually exclusive and reflect different mechanisms of aberrant social learning linked to paranoia.

Reviewer #2: The paper describe a computational analysis of previously collected behavioral data in a study about paranoia that used a modified serial dictator game. Using a generative Bayesian model, those new results demonstrate how paranoia may be associated with uncertainty about other’s action, moderate the relationship between learning rates and harmful intent attributions, and how the intention attributions are related to their precision.

This is a timely piece combining computational psychiatry, social interactive task, and network analyses. The text is pleasant to read thanks to clear hypotheses and interpretations. It is also great to see both pre-registration and open source code. Since the manuscript is quite dense, it would just benefit from some clarification, especially in the method section. Overall, the article should meet the standard for publication in PLoS Computational Biology after some minor revisions.

Main points

- The causal claims like “underly” (in the title) or “predict” (in the text) may be tempered since there is no empirical demonstration about the underlying causal mechanisms of paranoïa.

- It is difficult to conclude about a “domain-general alteration” since there is no “non-social” context tested in the current study. To what extend paranoïa could also apply to harmful catastrophe (e.g. earthquakes)?

- The “Bayesian Brain” needs to be introduced more clearly. There is a common map-territory fallacy in the domain when papers apply Bayesian methods and simultaneously support a Bayesian perspective on what the brain does. The notion of “engram encoding probability distributions” should be especially more detailed.

Minor points

- The sentence at lines 39-43 is hard to process.

- The modification of the dictator game is not explicitly described, even in the methods (lines 383-393).

- Sometimes the term “Participants” is used in the simulation context and it must be clear when we are talking about “Virtual Participants” or “Real Participants” (e.g. Fig 2 line 169).

- The use of acronyms make the piece somehow hard to follow. e.g. “GPTS” is defined only in the methods (line 368) and could easily be introduced in the main body of the text, e.g. line 88-94 with “pre-existing paranoid belief (as measured by the GPTS questionnaire)” or on those lines. This would drastically facilitate readers understanding. Same with parameters acronyms, which makes sense only after reading the methods. e.g. “pHI0” could be replace by “the initial probability of Harmful Intent (pHI0)” at first so that reader can get familiar with the notations.

- Statistics

- Line 110: What can be concluded by the fact that log likelihood did not drop below -4.394? For people not familiar with Bayesian statistics, it may be nice to just briefly explain the motivation to report this.

- Lines 111-115 / Lines 191-193: How can we interpret that as trials progress and GPTS increase, model fit is worse? To what extend the correlation between model fit and experimental factors is not problematic?

- Lines 223-234: This paragraph should more clearly written, especially regarding the cluster definition and rational of the analysis.

- Difference between “pairwise” and “moderation” should be better explained.

- What are the grid search parameters for the MAP on (line 492)?

- Figure

- 1 & 2: X axis labels have missing spaces

- 5: Doesn't left panel have CI or are they too small?

Reviewer #3: Barnby and colleagues report their efforts to model attributions of intent arising during a series of dictator games, across the spectrum of paranoia in healthy people. They claim that the model recapitulates their behavioral results and that their analyses favor a specifically social interpretation of paranoia.

I am afraid I think this is a stretch and I feel more unpacking - and possibly further experiments - are necessary before sharing such a conclusion.

1) There are social/non-social behavioral manipulations in which people with higher paranoia have been shown to have more inflexible social beliefs without the need for modeling - eg Wellstein et al Schiz. Res. 2019. How do the present results compare to that work? It seems inappropriate to call the selfish attribusion non-social in the context of the current experiment and modeling. Can the authors unpack their reasoning here?

2) 18 trials (comprised of six trials from each of three dictators) is extremely low. Can the authors show that their parameter estimates are stable and reliable?

3) The table describing the meanings of parameter estimates is a welcome inclusion, Unfortunately it is rather poorly written. It does emphasize that there are quite a few parameters in the models (almost as many as there are conditions). What are the effects on estimates/reproduction of behavioral features if some of those key parameters are removed?

4) The control analysis for the Network exploration is also a very good idea. Unfortunately, the authors seem to show that their fitting procedure is generating relationships between parameters. I suggest on that basis that the network analysis results be removed.

5) More clarity on the behavioral features from Barmby et al (2020) that they are seeking to recapitulate with the model would be useful. What behavioral features should the model produce? How exactly does it produce them?

6) Order effects of the dictators? Each human participant saw a fair, partially fair, and an unfair dictator in a random order. However, the model cares about these transitions, and, one would imagine, so do real participants. How well does the model capture the generalization from one dictator to the next? Does interacting with an unfair dictator first particularly impact highly paranoid people? Can the model capture that? It would seem that such an effect would be key to the authors interpretation of their prior published work, and it would be critical therefore that a learning model should capture this (such transitions from fair to unfair, for example, might bring these data in line with other approaches - like those of Nour, Wellstein, Reed - wherein volatility and participants responses to it seems to be key to paranoia - without such an analysis, I am not sure the present analysis has much to say about that other work and neither favorable nor unfavorable comparisons to that work are appropriate.

7) Parameter recovery. The simulated data produce simulated behavioral ratings. If the authors fit their model back to those simulated data what is the relationship between the estimates from real behavioral data and simulated data?

Reviewer #4: Review uploaded as an attachment.

**Have all data underlying the figures and results presented in the manuscript been provided?**

Reviewer #1: Yes

Reviewer #2: Yes

Reviewer #3: Yes

Reviewer #4: Yes

PLOS authors have the option to publish the peer review history of their article (what does this mean?). If published, this will include your full peer review and any attached files.

Reviewer #1: No

Reviewer #2: Yes: Guillaume Dumas

Reviewer #3: No

Reviewer #4: No
---

## [Decision Letter · Decision Letter 1]

7 Sep 2020

Dear Mr Barnby,

We are pleased to inform you that your manuscript 'Reduction in social learning and increased policy uncertainty about harmful intent is associated with pre-existing paranoid beliefs: evidence from modelling a modified serial dictator game.' has been provisionally accepted for publication in PLOS Computational Biology.

Best regards,

Daniele Marinazzo

Deputy Editor

PLOS Computational Biology

Daniele Marinazzo

Deputy Editor

PLOS Computational Biology

Reviewer's Responses to Questions

**Comments to the Authors:**

Reviewer #2: The authors have convincingly and clearly addressed all the points raised in my previous review.

I therefore endorse the publication of their manuscript in its current form.

**Have all data underlying the figures and results presented in the manuscript been provided?**

Reviewer #2: Yes

PLOS authors have the option to publish the peer review history of their article (what does this mean?). If published, this will include your full peer review and any attached files.

Reviewer #2: **Yes: **Guillaume Dumas

---

## [Editor Report · Acceptance letter]

7 Oct 2020

PCOMPBIOL-D-20-00671R1 

Reduction in social learning and increased policy uncertainty about harmful intent is associated with pre-existing paranoid beliefs: evidence from modelling a modified serial dictator game.

Dear Dr Barnby,

I am pleased to inform you that your manuscript has been formally accepted for publication in PLOS Computational Biology. Your manuscript is now with our production department and you will be notified of the publication date in due course.

With kind regards,

Matt Lyles
